# ERNIE-UIE: Advancing information extraction in Chinese medical knowledge graph

**Bei Li**[1¤a], **Changbiao Li**[1¤a], **Jianwei Sun**[2¤b], **Xu Zeng**[2¤b], **Xiaofan Chen**[1¤a], **Jing Zheng**[2*¤b]

1 Department of Biomedical Informatics, School of Life Science, Central South University, Changsha, Hunan, China, 2 Shenzhen Health Development Research and Data Management Center, Shenzhen, Guangdong, China

¤a Current Address: Department of Biomedical Informatics, School of Life Science, Central South University, Changsha, Hunan, China
¤b Current Address: Shenzhen Health Development Research and Data Management Center, Shenzhen, Guangdong, China
* jingcheung_silence@outlook.com

## Abstract

### Background

The field of information extraction (IE) is currently exploring more versatile and efficient methods for minimization of reliance on extensive annotated datasets and integration of knowledge across tasks and domains.

### Objective

We aim to evaluate and refine the application of the universal IE (UIE) technology in the field of Chinese medical expertise in terms of processing accuracy and efficiency.

### Methods

Our model integrates ontology modeling, web scraping, UIE, fine-tuning strategies, and graph databases, thereby covering knowledge modeling, extraction, and storage techniques. The Enhanced Representation through Knowledge Integration-UIE (ERNIE-UIE) model is fine-tuned and optimized using a small amount of annotated data. A medical knowledge graph is then constructed, followed by validating the graph and conducting knowledge mining on the data stored within it.

### Results

Incorporating the characteristics of whole-course management, we implemented a comprehensive medical knowledge graph–construction model and methodology. Entities and relationships were jointly extracted using the pretrained language model, resulting in 8,525 entity data points and 9,522 triple data points. The accuracy of the knowledge graph was verified using graph algorithms.

**Data availability statement:** All relevant data are within the manuscript and its Supporting Information files.

**Funding:** This work was supported by the Shenzhen Health Development Research and Data Management Center(grant number 738020018) and the first batch of key research topics on medical and health system reform of China Health Economics Society. The funders had important role in the design of the study; in the writing of the manuscript, or in the decision to publish the results.

**Competing interests:** The authors have declared that no competing interests exist.

## Conclusion

We optimized the construction process of a Chinese medical knowledge graph with minimal annotated data by utilizing a generative extraction paradigm, validating the graph's efficacy and achieving commendable results. This approach addresses the challenge of insufficient annotated training corpora in low-resource knowledge graph construction, thereby contributing to cost savings in the development of knowledge graphs.

---

## 1. Introduction

Information extraction (IE) is a transformative process that converts unstructured text into structured data. When faced with numerous intricate and diverse IE tasks, we often find ourselves developing various IE models to cater to the multifaceted requirements of these complex tasks. In the field of medical knowledge graph research, knowledge extraction primarily covers entity extraction and relationship extraction. Medical entity extraction, also referred to as named entity recognition, can be broadly classified into three main methodologies, namely, rule-based, statistical model-based, and deep learning-based methods [1]. Each method presents its own application scenarios, along with distinct advantages and limitations, and multiple methods may be combined in practical applications so as to enhance the entity recognition performance. For instance, Liu et al. conducted entity recognition based on a semisupervised learning framework utilizing a K-nearest neighbor classifier and a conditional random fields (CRF) model to effectively address the challenge of insufficient training data, thereby notably improving the performance of entity recognition [2].

In the medical domain, entity-relation extraction particularly refers to the process of identification of medical entities from unstructured medical texts, and determination of relationships between these entity pairs on the basis of predefined relationship types [3]. Existing relation-extraction methods can be broadly categorized into four types: (1) traditional rule-based and pattern-matching methods; (2) supervised machine learning methods, such as support vector machines [4], CRF [5], convolutional neural networks, and graph neural networks [6,7]; (3) semisupervised and unsupervised methods, such as bootstrapping [8] and k-means clustering algorithms; and (4) remote supervision methods [9]. With the rapid advancement of deep learning, neural network-based relation-extraction methods have increasingly become the focus of research, significantly enhancing the performance of relation-extraction. For instance, bidirectional encoder representations from transformers (BERT) and its variants, BioBERT and Clinical BERT, which are extensively used in medical relation-extraction research, are constructed based on neural networks [10]. These language models are superior to other machine learning and deep learning methods in that they do not require feature generation and representation and allow for the entire text to be used as model input, enabling learning from context. In the field of materials science, Hei et al. proposed a novel framework integrating pointer networks and augmented attention mechanisms to extract complex multi-tuple relations from scientific

literature, particularly those pertaining to alloy mechanical properties [11]. Their approach was designed to overcome the limitations of traditional methods in handling nested entities, contextual ambiguity, and one-to-many relational mappings. However, unlike materials science, medical informatics presents additional challenges, including higher annotation costs and a more dynamic data landscape—particularly in response to emerging disease variants. These characteristics necessitate the development of models with enhanced transfer learning capabilities. To address continual relation classification, Pang et al. introduced a strategy that combines knowledge distillation with contrastive learning to mitigate the problem of catastrophic forgetting. While effective in general domains, their method did not account for the unique temporal dynamics of medical data, such as the frequent updates to clinical guidelines and evolving diagnostic criteria [12].

In the development of Chinese medical information extraction, scholars have made continuous progress while simultaneously uncovering a range of challenges unique to this domain. One major linguistic obstacle arises from the absence of natural word boundaries in the Chinese language, which frequently leads to incorrect segmentation of clinical terms during the tokenization stage. Such segmentation errors can propagate through downstream tasks—most notably named entity recognition—resulting in cascading inaccuracies. For example, the phrase "高血压病史三年" (a three-year history of hypertension) should be recognized as a single unit representing a medical condition. However, if it is incorrectly segmented into "高/血压/病史" (high/ blood pressure/ history), the semantic integrity of the expression is easily compromised. Moreover, in contrast to English-language biomedical text mining—which benefits from well-established ontologies such as UMLS and SNOMED CT—the Chinese context still largely relies on manually curated lexicons or domain-specific knowledge graphs, such as CMKG [13]. These resources often exhibit limitations in terms of coverage, connectivity, and the degree of structural formalization. In addition, the sensitive and regulated nature of medical data presents a further obstacle, significantly restricting access to high-quality, publicly available corpora and annotated samples. Consequently, Chinese medical information extraction faces multifaceted challenges spanning linguistic representation, knowledge modeling, and task-level robustness.

With the rapid advancement of large language models (LLMs), the landscape of information extraction is undergoing a profound transformation. Unlike traditional approaches based on classification or sequence labeling, LLMs have ushered in a new paradigm known as generative information extraction, wherein extraction tasks are reformulated as text generation problems. This paradigm enables the model to produce structured outputs—such as knowledge triples—in natural language form [14]. Wadhwa et al. suggest that in the era of LLMs, relation extraction is shifting from typological modeling toward open-domain, question-answering-based modeling. By leveraging natural language prompts, this approach enhances scalability and cross-domain transferability [15]. Nevertheless, generative extraction approaches continue to grapple with challenges such as inconsistent outputs and the omission of multi-entity or complex relational structures. To address these limitations, recent studies have advocated for the use of predefined schemas or domain ontologies to guide model outputs, thereby enforcing consistency and completeness. For instance, schema-conditioned extraction frameworks utilize structured templates to constrain and inform generation, offering a promising pathway toward building high-quality, domain-specific knowledge graphs [16].

At present, prevailing knowledge extraction methods still rely heavily on extensive annotated datasets to train models, whereby the mapping between input data and corresponding output labels is learned to facilitate accurate information extraction. With advancements in pretraining techniques and transfer learning, the field of knowledge extraction has begun to seek methods for constructing more versatile, efficient, and flexible knowledge-extraction models, to explore whether cross-task and cross-domain knowledge integration can be quickly achieved, and reduce the dependence on large-scale labeled data. For instance, Google's Text-to-Text Transfer Transformer (T5) model exemplifies a direct text-to-text transformation method, unifying all natural language processing (NLP) tasks into text-to-text problems. Likewise, Baidu's Enhanced Representation through Knowledge Integration-universal IE (ERNIE-UIE) employs a unified framework to handle various types of IE tasks, thereby enabling the sharing of knowledge and capabilities across different tasks. The end-to-end learning capabilities of these methods further simplifies the extraction process by directly generating structured

outputs from text, thereby reducing reliance on complex pipelines and multistep processing. These methods represent a shift in the domain of knowledge-extraction research, from specialized, resource-intensive solutions to more general, data-efficient methodologies.

This study presents empirical research and discussion of the ERNIE-UIE technology, newly proposed by the Chinese Academy of Sciences and Baidu Inc., providing a detailed pathway and a low-cost, high-efficiency implementation scheme for the extraction of Chinese medical knowledge and construction of knowledge graphs. Building upon the ERNIE-UIE framework, this study incorporates a schema grounded in established medical ontologies to address domain-specific challenges inherent in medical information extraction. To enhance the coverage and diversity of input data, medical texts were expanded through automated web-crawling techniques. Furthermore, a unique relation-mapping strategy was introduced to resolve instances where ERNIE-UIE fails to generate complete relational tuples, particularly with missing tail entities. This work establishes a transferable and systematic methodology that extends the utility of ERNIE-UIE-based models for medical knowledge extraction, knowledge graph construction, and downstream clinical applications.

## 2.  Related technologies

UIE is an innovative IE technology introduced by Lu et al. at Association for Computational Linguistics 2022. Derived from the T5 model, it employs a unified structure generation framework to perform the unified modeling of diverse IE tasks [17].

The T5 model is based on the idea of considering all NLP tasks as "text-to-text" transformation tasks [18]. This allows for a unified and streamlined method to handle various NLP tasks by internally framing them as a single problem: "Given this input text, predict the subsequent text." Based on the transformer architecture, the T5 model employs a transfer learning strategy, wherein it is pretrained on a vast amount of unlabeled data and subsequently fine-tuned on specific downstream tasks. The "text-to-text" transformation concept of the T5 model has profoundly influenced the UIE model, which can be considered an extension of the T5 model. For the UIE model, whether it is entity recognition, relation extraction, or event extraction, these IE tasks can generate output text based on the input text. For instance, an entity recognition task can be regarded as follows: given a sentence that contains entities (input text), generate a new sentence annotated with the identified entities (output text).

To facilitate task switching, the UIE model incorporates the concept of prompts—a technique used in NLP tasks to guide the model in generating specific outputs [19]. By unifying various IE tasks into an input format of "Prompt + Text," with different prompts representing different tasks, the UIE model can produce diverse extraction results. In the UIE model, the construction of a prompt involves two essential elements: spotting and associating. Prompts built on these two elements are called structural schema instructors (SSI), essentially serving as a schema-based prompt mechanism for controlling various generation requirements. For example, the prompt format for entity extraction tasks is "[spot] entity type [text]," while that for relation-extraction tasks is "[spot] entity type [asso] relation type [text]." When SSI and text are input into the UIE model, the output is in structured extraction language, which contains extraction results in a specific format for users to parse and extract (Fig 1).

The following are the contributions of this study:

(1) As is known, Baidu's open-source ERNIE-UIE is the first Chinese UIE model. This study fully combines this model's high accuracy and efficiency in processing of Chinese text to propose a scheme that enables efficient and accurate knowledge extraction with a small amount of annotated data. This scheme can be easily adapted to various domains by using the trained model and predefined schemas, significantly enhancing the efficiency of knowledge graph construction.

(2) Incorporating the characteristics of whole-course management, we implement a comprehensive medical knowledge-graph–construction model and methodology. Our model integrates ontology modeling, web scraping, UIE, fine-tuning strategies, and graph databases, thereby covering knowledge modeling, extraction, and storage techniques.

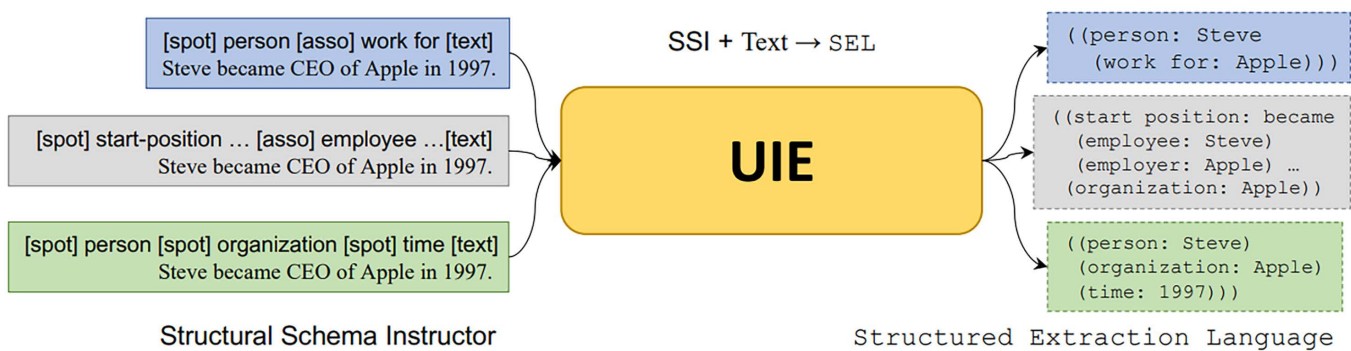

**Fig 1. Workflow of universal information extraction (UIE) [17].**

## 3. Materials and methods

### 3.1. Overall framework

A knowledge graph can be broadly divided into the pattern layer and data layer, as shown in Fig 2. The pattern layer refers to the ontology model, providing the foundational framework for the subsequent joint extraction of entities and relations in the process of knowledge extraction. The construction of the data layer, guided by the pattern layer, involves the acquisition of knowledge triples (<head entity, relationship, tail entity>) that conform to the framework definition.

All data used in this study were obtained from publicly available sources. Web scraping was conducted strictly in accordance with the robots.txt protocols of the respective websites, ensuring compliance with their terms of service. Literature-derived data were exclusively sourced from open-access publications and public domain resources. The data collection and analysis methods fully complied with the terms and conditions of the original data sources. Careful attention was paid to respecting intellectual property rights and avoiding any potential copyright infringements. No proprietary or restricted-access data were utilized in this research. The data acquisition and use processes were designed to maintain ethical standards and legal requirements for scientific research. More than 260,000 words of text data covering the prevention, diagnosis, treatment, and rehabilitation stages of the whole-disease-course management of hypertensive intracerebral hemorrhage were obtained through official documents, textbooks, literature, and medical website data. Using existing ontologies such as the Omaha Schema and the hypertension ontology within the Open Biological and Biomedical Ontology framework as a reference, an ontology model for hypertensive intracerebral hemorrhage was constructed, based on which the entity and relationship types were determined, following which the text data were labeled. After converting the labeled data into training data, the ERNIE-UIE model was fine-tuned as the subsequent knowledge-extraction model. Finally, the collected text data were input into the fine-tuned model so as to obtain triples, which were then processed and stored in the Neo4j graph database after manual proofreading. A hypertensive intracerebral hemorrhage knowledge graph was thus obtained.

### 3.2. Data preprocessing

The data sources used in the study were obtained via crawling the question-and-answer data from the professional medical Q&A website Dingxiang Doctor, combined with raw text data from professional literature and books sourced from China National Knowledge Infrastructure, Wanfang Medical Network, and WeChat Reading, totaling 261,494 characters.

The ERNIE-UIE model exhibits several features. First, during extraction, it constructs a prompt as an input to facilitate the extraction of triples. The prompt is actually a structure consisting of the head entity and the relationship of the triple in the form of "A's B" or "B" which conforms to natural language semantics, such as "disease's required measures." Second,

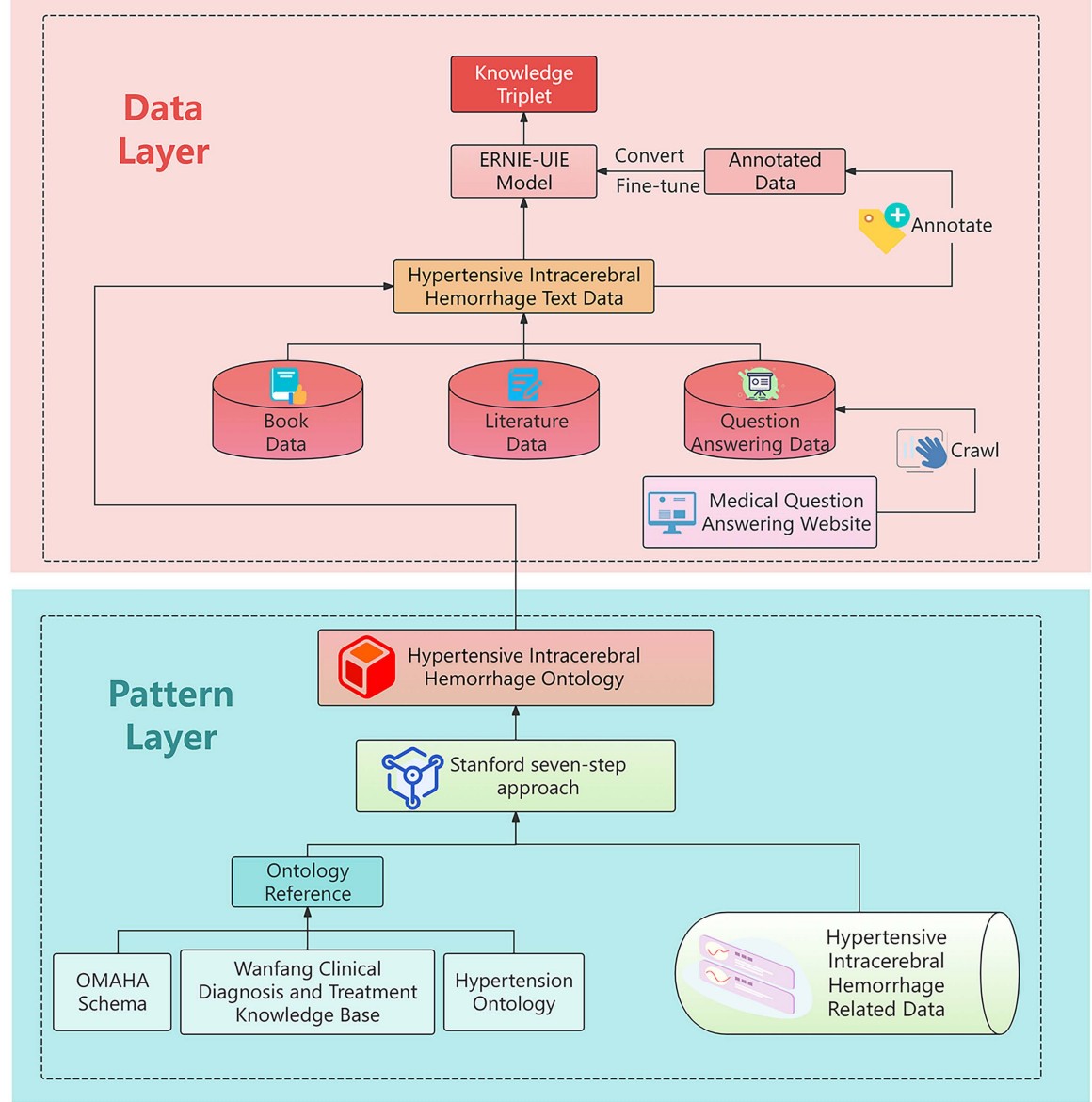

**Fig 2. Construction of the pattern layer and data layer of a knowledge graph.**

the JSON files acquired by this model's extraction process contain the specific content of the head entity, entity type of the head entity, relationship type, and specific content of the tail entity. However, they lack information regarding the entity type of the tail entity.

Therefore, the eight predefined relationship types (i.e., object properties) were refined by name so as to obtain 14 relationship types, ensuring that the prompt constructed in subsequent work had certain semantics. In addition, a unique relationship-type strategy was adopted to address the problem of missing tail entity-type information in the extraction results. This strategy allowed for the mapping of extraction results on the basis of relationship types so as to infer the entity type of the tail entity. The refined entity-relationship types are presented in Table 1, in which each relationship type

**Table 1. Entity-relationship types.**

| Head Entity Type | Relationship Type | Tail Entity Type |
| --- | --- | --- |
| Disease | Has symptom | Symptom |
| Disease | Requires examination | Examination |
| Disease | Requires measure | Other Measures |
| Disease | Involves department | Department |
| Disease | Leads to | Disease |
| Disease | Involves equipment | Equipment |
| Disease | Involves diet | Diet |
| Examination | Involves vital sign | Vital Sign |
| Surgical Procedure | Treats disease | Disease |
| Person | Is responsible for | Other Measures |
| Drug | Treats | Disease |
| Drug | Alleviates | Symptom |
| Other Measures | Involves drug | Drug |
| Other Measures | Has effect | Effect |

is unique and so is its corresponding tail entity type. Thus, the extracted relationship type can be used to infer the missing tail entity-type information. For instance, if the data extracted are "Disease | hypertensive intracerebral hemorrhage | involves equipment | suction device," where "Disease" is the entity type of the head entity "hypertensive intracerebral hemorrhage" and "involves equipment" the relationship type, referring to Table 1, then the entity type of the tail entity "suction device" is identified as "equipment." The refined data thus become "Disease | hypertensive intracerebral hemorrhage | involves equipment | Equipment | suction device."

**(1) Data annotation.** Based on the fine-tuning strategy for the ERNIE-UIE model, a small amount of annotated data was needed for subsequent fine-tuning of the model. From the text data collected, 200 entries (with each line in a txt document representing one data entry) were extracted and annotated using Doccano (a document annotation tool) (Fig 3).

The data obtained after manual annotation performed using Doccano were stored in JSON files. The content recorded included the text ID, text content (text), entity ID, entity type (label), starting and ending positions of the entity in the original text (start_offset and end_offset), relationship ID, relationship type (type), and corresponding head entity ID (from_id) and tail entity ID (to_id).

**(2) Data conversion.** The data annotated using Doccano (Fig 4) could not be directly used for model training and needed to be processed into a suitable format. The data were converted using the official Paddle Doccano annotation file conversion script, as shown in Fig 5. In the data converted, "content" corresponded to the original "text," while "result_list" denoted a list that extracted the specific content of the tail entities from the original annotated data. In addition, "start" and "end" corresponded to the "start_offset" and "end_offset" in the original annotations. The relationship types from the original annotated data were combined with the head entities so as to form prompts in the format of "head entity's relationship type." For example, in the prompt "脑出血的具有症状" ("intracerebral hemorrhage has symptom") in Fig 5, "脑出血" ("intracerebral hemorrhage") was the head entity and "具有症状" ("has symptom") the relationship type involved in these data.

## 4. Results

### 4.1. Model selection

The model employed in this study is ERNIE-UIE, a unified information extraction framework developed based on UIE by PaddleNLP. It utilizes ERNIE 3.0—a knowledge-enhanced large-scale language model within the Wenxin (文心)

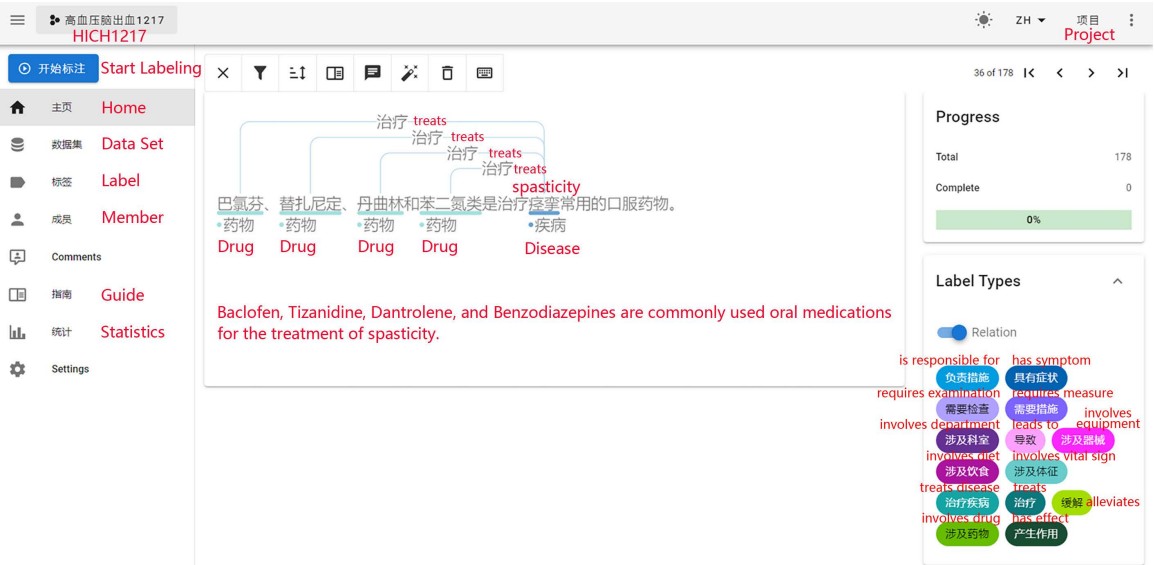

**Fig 3. Standard interface of Doccano.**

```
{"id":1152,"text":"If often sudden onset, severe headache, vomiting, hemiplegia, and disturbance of
 consciousness occur, cerebral hemorrhage should be considered.","entities":[{"id":3506,
 "label":"Symptom","start_offset":8,"end_offset":12},{"id":3507,"label":"Symptom","start_offset":13,
 "end_offset":15},{"id":3508,"label":"Symptom","start_offset":16,"end_offset":18},{"id":3509,
 "label":"Symptom","start_offset":19,"end_offset":23},{"id":3510,"label":"Disease","start_offset":28,
 "end_offset":31}],"relations":[{"id":1355,"from_id":3510,"to_id":3509,"type":"has symptom"},
 {"id":1356,"from_id":3510,"to_id":3508,"type":"has symptom"},{"id":1357,"from_id":3510,"to_id":3507,
 "type":"has symptom"},{"id":1358,"from_id":3510,"to_id":3506,"type":"has symptom"}],"Comments":[]}
```

**Fig 4. Example of original annotated data.**

```
{"content": "If often sudden onset, severe headache, vomiting, hemiplegia, and
 disturbance of consciousness occur, cerebral hemorrhage should be considered.",
 "result_list": [{"text": "disturbance of consciousness", "start": 19, "end": 23},
 {"text": "hemiplegia", "start": 16, "end": 18}, {"text": "vomiting", "start": 13,
 "end": 15}, {"text": "severe headache", "start": 8, "end": 12}], "prompt": "Symptoms
 of cerebral hemorrhage"}
```

**Fig 5. Example of converted data.**

foundation model family—as its encoder, and adopts a dual-pointer decoding architecture. ERNIE-UIE is a general-purpose Chinese information extraction model trained through large-scale multi-task learning. By leveraging universal extraction capabilities acquired via extensive multitask pretraining, alongside a prompt-based unified modeling approach for information extraction tasks, ERNIE-UIE demonstrates robust adaptability across a wide array of domains and extraction objectives. Notably, the model supports zero-shot initialization, enabling rapid deployment without prior task-specific annotation. Furthermore, it exhibits strong few-shot fine-tuning performance, achieving substantial

improvements in domain-specific information extraction even with limited annotated data. The experiments were conducted using the PaddlePaddle AI Studio environment, with a GPU configuration of Tesla V100 and a video memory of 16 GB. The distributed deep learning framework PaddlePaddle and Baidu's PaddleNLP library for NLP were employed to jointly extract entities and relationships on the basis of UIE. The versions of PaddlePaddle and Paddle NLP were paddlepaddle-gpu 2.6.0 and PaddleNLP 2.6.1, respectively.

**(1) Comparison of ERNIE-UIE models.** The ERNIE-UIE framework offers multiple UIE models with various structures for selection. Herein, we compared the "uie-nano," "uie-base," and "uie-m-large" models, all of which support Chinese IE, to identify the optimal model as the final knowledge-extraction model.

The models were trained and evaluated on the same annotated dataset, with performance assessed across three metrics: Precision, Recall, and F1-score. The training results of the three models are presented in Table 2. The "uie-base" model delivered a superior overall performance and was selected as the final model for fine-tuning and subsequent knowledge extraction in this study.

**(2) Experimental results.** The ratio of the training set to testing set was 8:2 in the experiment. The training parameters were set as follows: Learning_rate was set to $1e^{-5}$, Batch_size to 16, and Epochs to 30. The extraction performances for entities and relationships are presented in Tables 3 and 4, respectively.

ERNIE-UIE delivered superior extraction performance for commonly used entity types and relationship types, such as the entity type "Disease" and the relationship type "alleviates". This efficacy could be attributed to its foundation as a pretrained language model, which resulted in the capability to extract general information acquired from extensive datasets during pretraining. However, the model poorly performed when tasked with extraction of certain custom-defined entity types and relationship types. A representative example was the entity type "Other Measures," the entities of which typically included longer phrases or sentences and lacked distinctive syntactic or grammatical features. Upon sampling the extraction results, we observed that the extracted entities and relationships rarely had logical and factual errors, indicating

**Table 2. Comparison of training results of different models.**

| Model | Precision | Recall | F1-score |
|---|---|---|---|
| uie-nano | 0.4996 | 0.3755 | 0.4288 |
| uie-base | 0.4958 | 0.5841 | 0.5363 |
| uie-m-large | 0.4833 | 0.4971 | 0.4901 |

**Table 3. Performance of entity extraction.**

| Entity type | Precision | Recall | F1-score |
|---|---|---|---|
| Disease | 0.9007 | 0.8946 | 0.8976 |
| Surgical Procedure | 0.6667 | 0.7692 | 0.7143 |
| Person | 0.7701 | 0.8171 | 0.7929 |
| Department | 0.6000 | 0.8000 | 0.6857 |
| Examination | 0.5976 | 0.7101 | 0.6490 |
| Drug | 0.7875 | 0.8514 | 0.8182 |
| Other Measures | 0.5152 | 0.1809 | 0.2677 |
| Vital Sign | 0.5985 | 0.6752 | 0.6345 |
| Symptom | 0.7531 | 0.5398 | 0.6289 |
| Diet | 0.6629 | 0.6146 | 0.6378 |
| Equipment | 0.8333 | 0.7317 | 0.7792 |
| Effect | 0.7941 | 0.3803 | 0.5143 |

Table 4. Performance of relationship extraction.

| Relationship type | Precision | Recall | F1-score |
|---|---|---|---|
| Is responsible for | 0.7692 | 0.3571 | 0.4878 |
| Has symptom | 0.6250 | 0.5682 | 0.5952 |
| Requires examination | 0.4800 | 0.5217 | 0.5000 |
| Requires measure | 0.5807 | 0.3051 | 0.4000 |
| Involves department | 0.7500 | 1.0000 | 0.8571 |
| Leads to | 0.9615 | 0.4464 | 0.6098 |
| Involves equipment | 0.7222 | 0.5652 | 0.6342 |
| Involves diet | 0.7419 | 0.5610 | 0.6389 |
| Involves vital sign | 0.5833 | 0.4070 | 0.4795 |
| Treat disease | 0.5882 | 0.4651 | 0.5195 |
| Treats | 0.5714 | 0.2162 | 0.3137 |
| Alleviates | 0.7719 | 0.6027 | 0.6769 |
| Involves drug | 0.8462 | 0.6471 | 0.7333 |
| Has effect | 0.6154 | 0.2462 | 0.3517 |

that they generally met application requirements. Consequently, future work could focus on the use of this fine-tuned model as a basis for jointly extracting entities and relationships from the data collected.

## 4.2. Joint entity-relationship extraction

The joint entity-relationship extraction method is an end-to-end model that simultaneously recognizes entities and extracts relationships between them. This method more effectively leverages the implicit relationships between entity types and relationship types, thereby enhancing the performance of relationship extraction. In general, these methods are based on neural networks, enabling the direct output of entity and relation combinations. Following the four stages of whole-disease-course management of prevention, diagnosis, treatment, and rehabilitation, the fine-tuned model was employed to jointly extract entities and relationships batchwise. Prior to the extraction process, it was essential to define the extraction framework (schema) with reference to predefined entity types and relationship types. In addition, the task type (in this study, the IE task "information_extraction") and the path to the trained model data (task_path) needed to be specified. After these parameters were set, we passed in the text, as a parameter, for extraction (text) to the "my_ie" function, and entities and relationships could be jointly extracted from the text as per the schema (Related codes are shown in S1 Appendix 1).

In this study, the ERNIE-UIE model was employed for knowledge extraction, and several key characteristics of the model are noteworthy. First, the model adopts a prompt-based strategy for triple extraction, in which the prompt serves as the input and is constructed using the head entity and the relation in a natural language-compliant format, such as "A's B" or the genitive form "B of A"—for instance, "intervention for disease." This design enables the generated prompt to align closely with linguistic intuitiveness and semantic clarity. Second, the output of the model—formatted as a JSON file—contains detailed information about the head entity (including its content and entity type), the relation type, and the tail entity content. However, it lacks information regarding the entity type of the tail entity. To address these issues, the semantic compatibility between the head entity and the relation was carefully considered during prompt construction, particularly in terms of the noun–verb collocational semantics, to ensure that the generated prompts retain meaningful interpretation and thereby enhance extraction performance. In addition, to compensate for the absence of tail entity type annotations, a strategy based on unique relation types was implemented. This approach allows for the post hoc inference of the tail entity type by mapping the extracted results according to predefined head–relation–tail structure patterns, wherein each unique relation reliably corresponds to a specific tail entity type.

The extraction results included multiple nested lists and dictionaries, which needed to be parsed before being stored and analyzed. In particular, the parsing logic involved converting the extracted text into a JSON-formatted string (as shown in Fig 6), followed by determining the hierarchical level and the presence of relationship-type data to save entities, entity types, and triples.

The parsed and saved data, as shown in Fig 7, could be categorized into two types: (1) data that comprised only entities and types thereof, and (2) a triple that comprised head entities, head entity types, relationship types, and tail entities.

From the extraction results in Fig 6, it is evident that the data extracted by ERNIE-UIE did not include the entity type of the tail entity. A mapping process was required to obtain this information. In particular, on the basis of the entity

```
1   [{
2       'Disease': [{
3           'text': 'Intracranial hemorrhage',
4           'start': 10,
5           'end': 13,
6           'probability': 0.9954337178786474,
7           'relations': {
8               'has symptom': [{
9                   'text': 'Constipation',
10                  'start': 14,
11                  'end': 21,
12                  'probability': 0.5503956115805408
13              }],
14              'requires measure': [{
15                  'text': 'Acupoint Taping',
16                  'start': 0,
17                  'end': 4,
18                  'probability': 0.8503731662122043
19              }],
20              'leads to': [{
21                  'text': 'Slight bowel movement',
22                  'start': 39,
23                  'end': 43,
24                  'probability': 0.7625829607421934
25              }, {
26                  'text': 'Constipation',
27                  'start': 14,
28                  'end': 21,
29                  'probability': 0.8467483434417602
30              }]
31          }
32      }, {
33          'text': 'Cerebral hemorrhage',
34          'start': 10,
35          'end': 13,
36          'probability': 0.9954337178786474
37      }],
```

**Fig 6. Extraction results converted to JSON format (part).**

| Head Entity Type | Head Entity | Probability | Tail Entity | Probability | Relationship Type |
|---|---|---|---|---|---|
| Disease | HICH | 0.971859955 | | | |
| Disease | Stroke | 0.999927164 | | | |
| Department | Rehabilitation Center | 0.584714605 | | | |
| Person | Patient | 0.999295836 | | | |
| Person | Rehabilitation Physician or Doctor | 0.999797235 | | | |
| Other Measures | Change Unhealthy Lifestyle | 0.98217756 | | | |
| Person | Therapist | 0.999843842 | | | |
| Other Measures | Turn the Patient to the Side and Supine Positions | 0.685469025 | | | |
| Disease | HICH | 0.971859955 | CT | 0.996235349 | requires examination |
| Disease | HICH | 0.999809393 | Pulmonary Infection | 0.981862841 | leads to |
| Disease | HICH | 0.999727201 | Strict Blood Pressure Control | 0.658713623 | requires measure |
| Disease | HICH | 0.999820001 | Right Cerebral Falx Herniation | 0.948149722 | leads to |
| Surgical Procedure | Craniectomy | 0.999564456 | HICH | 0.997204821 | treat disease |
| Surgical Procedure | Tracheostomy | 0.998993497 | Aspirational and Atelectatic Pneumonia | 0.994013828 | treat disease |

**Fig 7. Example of parsed data.**

and relationship types defined in Table 1, the name of the relationship type in the triples extracted was used to infer its uniquely corresponding tail entity type, which was then added to the parsed results (related codes are provided in S2 Appendix 2). For instance, the mapping rule "relation_to_label = {"is responsible for": "Other measures","has symptom": "Symptom", "requires examination": "Examination", "requires measure": "Other Measures", "involves department": "Department", "leads to": "Disease", "involves equipment": "Equipment", "involves diet": "Diet", "involves vital sign": "Vital Sign", "treats disease": "Disease", "treats": "Disease", "alleviates": "Symptom", "involves drug": "Drug", "has effect": "Effect"}" illustrated the one-to-one correspondence between relationship types and tail entity types. If the relationship type in the triple was "is responsible for," the tail entity type must be "other measures." The entity type of the tail entity was obtained after deduplication through mapping. However, we also observed during data processing that a small portion of head entities exhibited nonunique entity types, meaning the same entity was recognized by the model as multiple entity types.

Separating this subset of entities and their corresponding entity types, we conducted manual cleaning, check, and verification to remove the "wrong entity type" data, obtaining a data table in which each entity was associated with a single entity type. After updating the original dataset via this part of corrected data, we rechecked all triple data as per the predefined entity and relationship types in Table 1 and eliminated any triples that did not conform to these definitions.

The final statistics of entities and relationships for each stage of whole-disease-course management are presented in Tables 5 and 6. Regarding the extraction results, the quantity distribution of the extracted entities or relationships aligned with the characteristic of whole-disease-course management of different stages with different focuses. For instance, the number of entities of the type"Surgical Procedure"was considerably higher during the treatment stage compared with other stages, while the number of relationships of the types "has symptom" and"requires examination"extracted in the diagnosis stage was significantly higher than that in other stages. An explanation may be that the diagnosis stage relies on symptoms and examinations to provide diagnostic conclusions. This indicated that the data extracted were reasonably distributed.

## 4.3. Knowledge storage of hypertensive intracerebral hemorrhage

Neo4j (a graph database) was the knowledge storage tool selected for this study. It is a high-performance graph database management system that employs graph theory-based storage structures to manage data. Developed by Neo Technology, it is one of the most popular graph databases on the market, particularly well-suited for handling highly interconnected data. Based on the graph data model, Neo4j's data structure consists of four components: nodes, relationships, properties, and labels. Compared with RDF storage, Neo4j offers significant advantages in terms of graph traversal

**Table 5. Statistics of entity extraction results.**

| Entity type | Prevention (N) | Diagnosis (N) | Treatment (N) | Rehabilitation (N) |
|---|---|---|---|---|
| Disease | 269 | 369 | 573 | 321 |
| Examination | 107 | 209 | 191 | 84 |
| Department | 9 | 9 | 25 | 17 |
| Other Measures | 397 | 44 | 770 | 793 |
| Equipment | 36 | 8 | 320 | 238 |
| Person | 217 | 48 | 123 | 174 |
| Vital Sign | 101 | 122 | 252 | 106 |
| Surgical Procedure | 8 | 13 | 282 | 22 |
| Drug | 173 | 19 | 286 | 134 |
| Diet | 716 | 2 | 55 | 82 |
| Symptom | 112 | 334 | 259 | 285 |
| Effect | 118 | 3 | 187 | 221 |

**Table 6. Statistics of relationship extraction results.**

| Relation type | Prevention (N) | Diagnosis (N) | Treatment (N) | Rehabilitation (N) |
|---|---|---|---|---|
| Has effect | 18 | / | 65 | 59 |
| Leads to | 279 | 423 | 582 | 288 |
| Is responsible for | 168 | 10 | 149 | 301 |
| Alleviates | 47 | 11 | 187 | 92 |
| Has symptom | 167 | 644 | 370 | 278 |
| Involves department | 16 | 17 | 70 | 21 |
| Involves equipment | 16 | 6 | 363 | 157 |
| Involves vital sign | 43 | 113 | 123 | 10 |
| Involves drug | 95 | 10 | 236 | 43 |
| Involves diet | 419 | 2 | 41 | 48 |
| Requires measure | 300 | 42 | 1107 | 372 |
| Requires examination | 140 | 642 | 320 | 91 |
| Treats | 63 | 4 | 141 | 45 |
| Treat disease | 9 | 13 | 380 | 22 |

performance, intuitive data modeling, real-time querying, and tool ecosystem, making it ideal for application scenarios requiring efficient graph operations and real-time responsiveness [20,21].

Neo4j version 5.11.0 was used for storage, and the py2neo library was employed to set up a route to connect to the local Neo4j graph database, thereby automatically importing relationship data and entity data in batches. During the creation of nodes, the "MERGE" form was employed to avoid duplication of the same entity. The knowledge graph was visualized using the Neo4j Browser, as shown in Fig 8.

Various types of knowledge regarding hypertensive intracerebral hemorrhage from the perspective of whole-disease-course management can be queried using the knowledge graph constructed. For instance, to query diet for the rehabilitation stage of hypertensive intracerebral hemorrhage, a Cypher query statement can be constructed as follows: "MATCH p = (d:disease {name: "hypertensive intracerebral hemorrhage"})-[:involves diet]- > (f:diet) WHERE "rehabilitation stage" IN d. related stages OR "rehabilitation stage" IN f. related stages RETURN p." The results, shown in Fig 9, indicate that

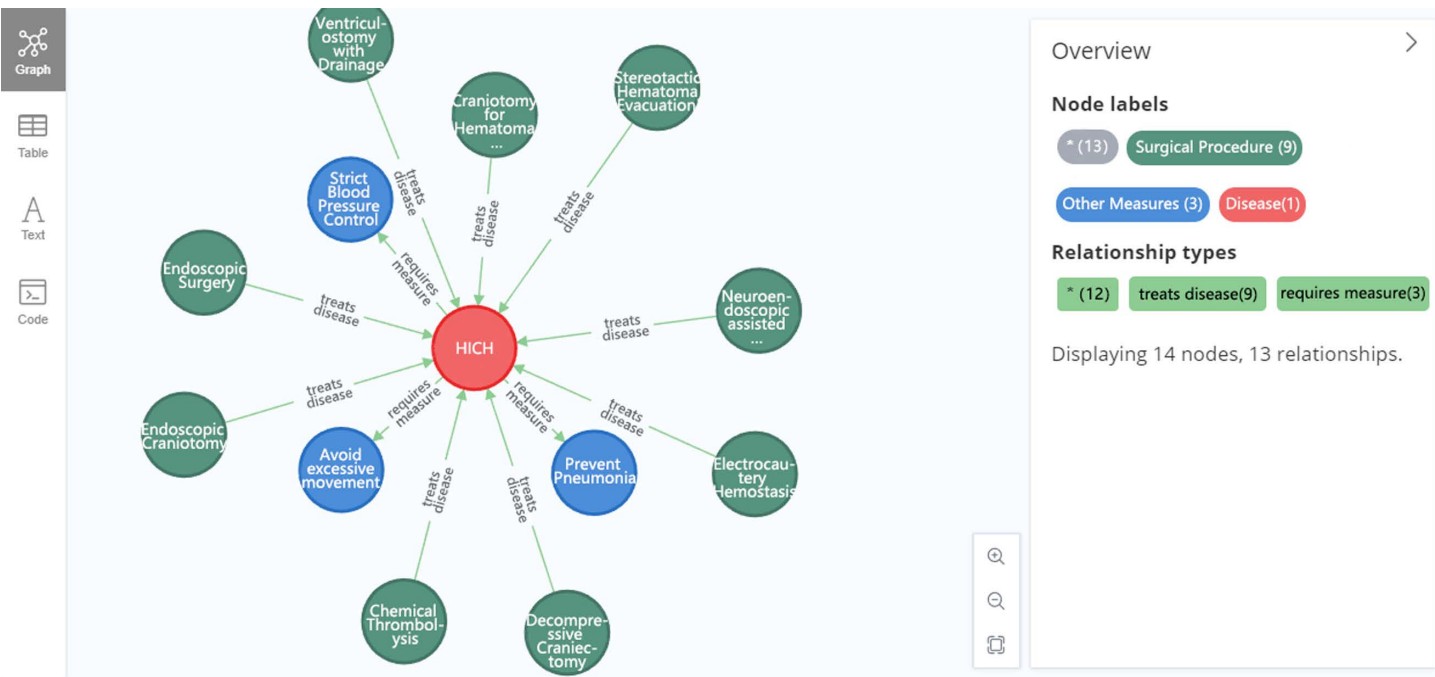

**Fig 8. Example of knowledge graph.**

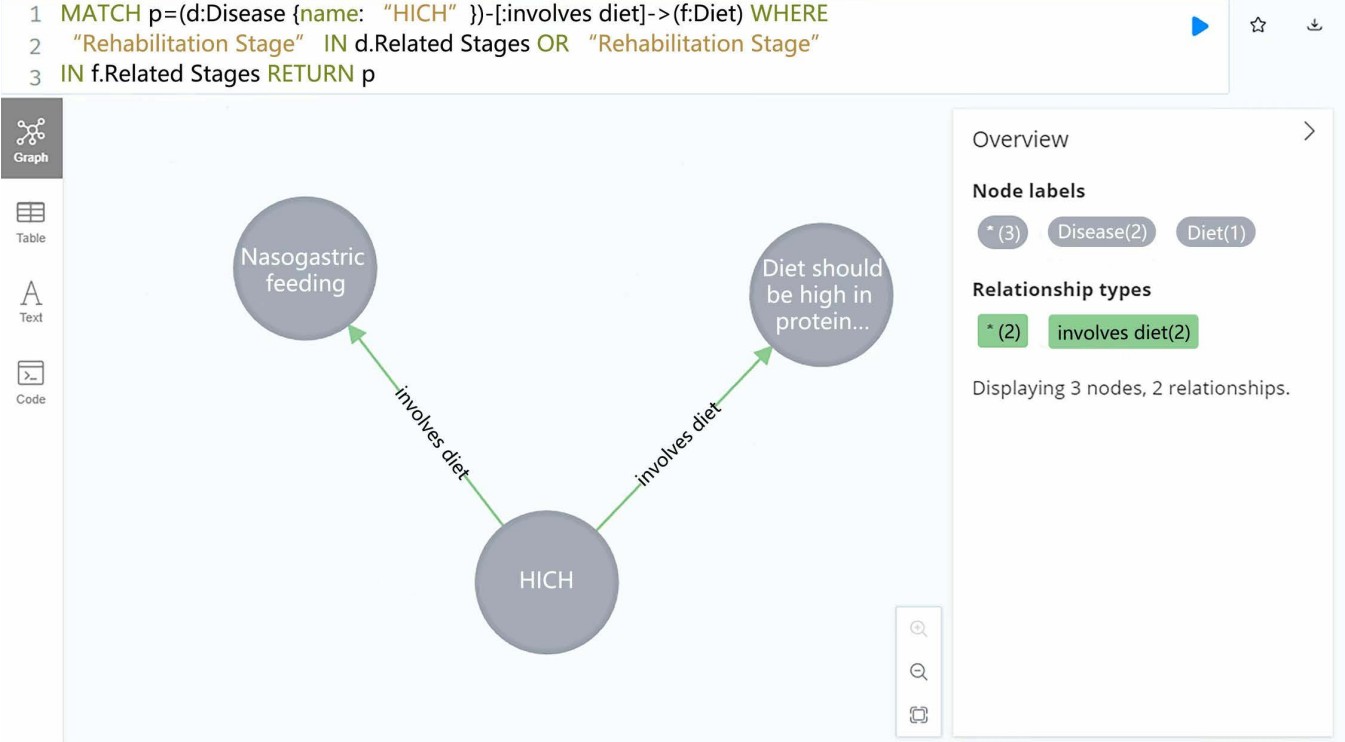

**Fig 9. Example of a knowledge query.**

during the rehabilitation stage of hypertensive intracerebral hemorrhage, (1) the diet should adhere to principles of high protein, high calorie, high vitamin, and low sugar and (2) nasal feeding may also be needed in special circumstances.

## 4.4. Knowledge graph validation and knowledge mining based on graph algorithms

After a knowledge graph is established, graph algorithms can be employed to process and analyze the information stored in the graph, thereby mining the hidden knowledge. Graph algorithms, which are based on graph theory and focus on handling graph data, utilize the relationships between nodes to infer the structure and changes in complex systems. These algorithms boast significant practical value and are widely employed across various fields, including social network analysis, e-commerce, bioinformatics, and network routing.

Using the graph algorithms provided in Neo4j, we validated the scientific nature of the knowledge graph and mined hidden knowledge stored within the graph. Graph data science was employed to conduct a series of analyses on an already constructed knowledge graph, with the aim of mining the knowledge hidden in the graph and validating its efficacy, thereby providing a foundation for subsequent applications. The Cypher query statement "CALL gds.graph.project('my-Graph,''*,''*')" was used to read and load the data and create a graph projection named "myGraph" for further analysis.

**(1) Centrality calculation.** The ArticleRank algorithm was employed to calculate the centrality, thereby determining the importance of different nodes within the network. This graph-based ranking method is a variant of the PageRank algorithm. While PageRank assumes that relationships originating from low-degree nodes (i.e., nodes with fewer connections) have greater influence, ArticleRank reduces the influence of these low-degree nodes by diminishing the score they transmit to their neighbors in each iteration. Using the Cypher query "CALL gds.articleRank.stream('myGraph') YIELD nodeId, score RETURN gds.util.asNode(nodeId).name AS name, score ORDER BY score DESC," we calculated the ArticleRank score for each node in the graph projection named "myGraph." The top ten nodes by ArticleRank score are shown in Fig 10.

"Hematoma" was the most crucial node within the entire knowledge graph. Hematoma, characterized by swelling or lump caused by clotted blood, serves as a critical diagnostic indicator in the diagnosis of hypertensive intracerebral hemorrhage. In general, the size of a hematoma is confirmed through a head CT scan and other examinations, based on which the selection is then made between conservative pharmacological treatment and surgical intervention so as to remove the hematoma. Thus, "hematoma" is a vital indicator influencing the diagnosis, treatment, and even rehabilitation of hypertensive intracerebral hemorrhage. Other nodes, such as "Hypertension", "Head CT", "Headache", "Antihypertensive Medication", "Antibiotics" and "DSA" also achieved high ArticleRank scores, whose significance aligned well with the existing knowledge on hypertensive intracerebral hemorrhage, indicating that the knowledge graph constructed had a certain degree of professionalism and credibility.

**(2) Topological link prediction.** We employed the classic common neighbors algorithm, which is a topological-link-prediction algorithm, to predict potential links between nodes in the graph. This algorithm is a network structure-based link prediction method, inspired by the intuitive observation in real life that two strangers with mutual friends are more likely to be introduced to each other. This algorithm offers several advantages: it effectively captures the community structure within the network; it requires only the topological structure of the graph, without needing additional attribute information of nodes or edges; and its prediction results are highly interpretable, with the prediction score directly representing the number of common neighbors (i.e., the shared neighbors) of two nodes. Using the Cypher query ""MATCH (e:disease {name: "hypertensive intracerebral hemorrhage"}), (a:other measures) WHERE NOT (e)-[:LINK]-(a) AND ("treatment stage" IN e. related stages OR "treatment stage" IN a. related stages) WITH e, a, gds.alpha.linkprediction. commonNeighbors(e, a) AS score RETURN e.name AS from, a.name AS to, score ORDER BY score DESC", we predicted "other measures" that may be required for "hypertensive intracerebral hemorrhage" during the treatment stage within the knowledge graph. The results are shown in Fig 11.

| name | score |
|------|-------|
| "Hematoma" | 3.1057085630489 |
| "Brainstem Hemorrhage" | 1.7524521586472228 |
| "Head CT" | 1.6937274342517068 |
| "Intracerebral hemorrhage" | 1.4262512527819875 |
| "Headache" | 1.3259477455307336 |
| "DSA" | 1.284174936488196 |
| "Hypertension" | 1.273394023647551 |
| "Antibiotics" | 1.1023760511653458 |
| "Antihypertensive Medication" | 1.0119053507921736 |
| "Hypertensive Intracerebral Hemorrhage" | 0.9937475399808211 |

**Fig 10. Top ten nodes by articlerank score.**

These prediction results were largely consistent with findings reported in the literature. For instance, during the treatment stage of "hypertensive intracerebral hemorrhage," "Maintain patent airways" is necessary because common complications of hypertensive intracerebral hemorrhage include pneumonia, aspiration, and respiratory failure or distress. It is essential to maintain airway patency to prevent these complications from occurring or to timely treat them. Measures such as "Intracranial pressure monitoring and ventricular drainage" and "Blood pressure management" are related to the frequent occurrence of hypertension during the acute phase of hypertensive intracerebral hemorrhage. These measures can effectively alleviate intracranial hypertension, improve treatment outcomes, and reduce mortality rates.

## 5. Discussion

Bonan Min et al. proposed generative unified modeling as one of the three mainstream paradigms in NLP today [22]. The predominant feature of mainstream IE modeling methods is that they require a given offset and perform tagging (position tagging), such as sequence tagging or span tagging. This extraction scheme heavily relies on extensive domain annotation data, which is extremely costly. There are often multiple IE requirements, such as entities and relationships in the same business, and also the cost of separate modeling and training is high. This method is called the "extraction paradigm." However, as generative pretrained models (e.g., the T5 model and bidirectional and autoregressive transformers model) become increasingly powerful, all NLP tasks can be transformed into the "text-to-text" format, also referred to as the "generative extraction paradigm." Use of generative pretrained models to address IE tasks can be termed as "paradigm shift" or "task reconfiguration."

| from   | to                                                       | score |
|--------|----------------------------------------------------------|-------|
| "HICH" | "Maintain patent airways"                                | 8.0   |
| "HICH" | "Endoscopic Hemostasis"                                   | 6.0   |
| "HICH" | "Intravenous Infusion"                                   | 6.0   |
| "HICH" | "Complete Bed Rest"                                       | 5.0   |
| "HICH" | "Relieve Pressure"                                        | 5.0   |
| "HICH" | "LP CSF Exchange Therapy"                                 | 5.0   |
| "HICH" | "Rest Quietly in Bed"                                     | 4.0   |
| "HICH" | "Intracranial Pressure Monitoring and Ventricular Drainage" | 4.0   |
| "HICH" | "Avoid Lower Extremity Intravenous Infusion"             | 4.0   |
| "HICH" | "Blood Pressure Management"                               | 4.0   |

**Fig 11. Example of link prediction results.**

In 2022, the Institute of Software of the Chinese Academy of Sciences, in collaboration with Baidu, introduced the "Chinese general information" extraction technology UIE, which unifies numerous tasks and lowers the threshold for engineering use of fundamental NLP tasks. This innovation swiftly attracted widespread attention within the industry. However, knowing whether this model can distinguish itself among various classic extraction techniques requires further validation. Currently, most application cases are focused on training on general-purpose corpora, such as those in legal and financial domains, wherein the model has demonstrated high accuracy. However, the medical domain, characterized by higher specificity, has fewer corpus instances and thus requires more data to support this model's efficiency and accuracy. Nevertheless, one cannot deny that UIE methods are exactly the "generative extraction paradigm," embodying the idea of end-to-end information extraction using a single model. This involves training on large corpora to acquire extensive semantic and syntactic knowledge, following which powerful pretrained models can be employed to accomplish diverse tasks in a generative manner. This method requires less labor, offers higher interpretability, and can more flexibly handle complex tasks. It also signifies a shift in NLP research works from specific domains to general model application.

In the field of knowledge graph construction, UIE models offer a more convenient and efficient approach for building of domain-specific knowledge graphs because of these models' inherent generality (the ERNIE-UIE model in this study), which enables them to handle IE tasks across various domains to acquire knowledge of entities, relationships, and attributes. Given that these models are built upon large pretrained models, they inherently possess strong capabilities for information comprehension and processing. Consequently, they perform well even in the extraction of complex entities

and relationships. In essence, knowledge-extraction methods based on the "generative extraction paradigm" address the challenge of lacking extensive annotated training corpora in low-resource knowledge graph construction and help reduce the construction cost of knowledge graphs, enhancing the efficiency of applications in the field of knowledge graph and bridging the gap between resource supply and demand.

In a multitask learning environment, the method of pretraining a model plus fine-tuning it offers distinct advantages—all tasks share the parameters of the pretrained model, possibly facilitating knowledge transfer between tasks and enhancing the model's performance across various tasks [23]. Therefore, our study leveraged a pre-established ontology model of hypertensive cerebral hemorrhage to collect and preprocess data. We fine-tuned the ERNIE-UIE model after acquiring a small amount of annotated data, thus achieving knowledge extraction in the medical domain with low cost and high efficiency. The model exhibited robust extraction capabilities for common entity types and relationship types. This method holds certain potential for application in the medical field. Drawing on our previous research [24], if it is applied to knowledge extraction in the medical domain, it can provide a reliable source of medical knowledge for medical question answering and the currently popular retrieval-augmented generation research.

However, the model poorly performed for certain user-defined entity types and relationship types (see Tables 3 and 4). This may be because it is a pretrained language model, because of which although it retains the capability to extract general information from large-scale data during pretraining, it is not ideal for extraction of long phrases or sentences that lack typical sentence patterns and grammatical features. After fine-tuning, some metrics continued to exhibit poor performance and there was no entity disambiguation for the knowledge extracted, resulting in some semantically similar entities within the final knowledge set.

Although the generative extraction paradigm utilized in this study demonstrated significant potential in knowledge graph construction and IE, its limitations cannot be ignored, such as its heavy reliance on the design of prompts for performance. Future research needs to address various aspects, including prompt engineering, knowledge updating, hallucination issues, complex-sentence processing, domain adaptability, multitask learning, computational resource requirements, entity disambiguation, few-shot learning, and interpretability, to further enhance the performance and practicality of the generative extraction paradigm. By integration of external knowledge sources, introduction of postprocessing steps, and designing of more sophisticated multitask learning frameworks, among other technical approaches, the generative extraction paradigm is poised to play an even greater role in the field of knowledge graphs [25,26].

Notably, in recent natural language processing (NLP) research, retrieval-augmented generation (RAG) has emerged as a widely adopted framework that integrates generative language models with external knowledge retrieval. This approach has demonstrated considerable utility across a range of complex tasks, including multi-hop question answering, information extraction, and knowledge graph construction. As the complexity of application scenarios continues to grow, research within the RAG paradigm has begun to shift from unstructured document retrieval toward more structured, controllable, and interpretable forms of knowledge integration. Of particular interest is the incorporation of graph-structured knowledge, as exemplified by recent frameworks such as GraphRAG and LightRAG. GraphRAG represents a notable advancement in this area by combining retrieval-augmented generation with knowledge graph reasoning. It transforms retrieved documents into entity-relation graphs and applies graph neural networks (GNNs) to enhance the model's capacity for structural semantic understanding and logical inference during text generation. Empirical evidence has shown that this method significantly improves answer consistency and factual accuracy on challenging multi-hop tasks such as HotpotQA [27].

However, GraphRAG faces practical challenges, including high graph construction overhead and deployment complexity. In response, lightweight alternatives such as LightRAG have been introduced. LightRAG simplifies the graph structure into shallow entity connectivity graphs and incorporates efficient retrieval indexing techniques, thereby maintaining comparable performance while substantially reducing inference latency and resource consumption [28]. To further advance the integration of generative models with structured information, recent studies have developed resources such as IEPile—a

schema-conditioned information extraction corpus designed to support transfer learning and capability evaluation of large language models in structured output tasks across multiple domains [16].

In addition, recent studies have explored the incorporation of graph-structured constraints directly into the decoding process of retrieval-augmented generation (RAG) frameworks. Recent studies have explored the direct incorporation of structural graph constraints into the decoding process of retrieval-augmented generation (RAG) frameworks. For instance, GraphRAFT—a retrieval and reasoning framework—fine-tunes large language models (LLMs) to generate verifiably correct Cypher queries, thereby retrieving high-quality subgraph contexts and enabling the production of accurate, evidence-based answers. This approach leverages the synergy between structured knowledge retrieval and generative reasoning, addressing both the reliability and precision of knowledge-intensive outputs [29]. In medical settings, certain approaches have sought to integrate specialized knowledge graphs to augment the performance of large language models (LLMs). For example, MedGraphRAG leverages a medical knowledge graph—customized specifically for processing private healthcare data—to enhance the capabilities of LLMs [30]. Collectively, the integration of graph-structured knowledge into RAG architectures has significantly enhanced the model's capabilities in structural recognition, factual reasoning, and semantic consistency. Future research may further investigate adaptive knowledge graph construction, cross-task representation sharing of graph structures, and interpretable graph-based inference. These directions hold substantial promise for expanding the utility of graph-enhanced RAG models in large-scale question answering and knowledge extraction tasks.

## 6. Conclusion

Based on Baidu's first open-source "Chinese general information" extraction model, ERNIE-UIE, we developed an efficient and accurate knowledge-extraction scheme requiring only a small amount of annotated data, fully leveraging the model's high accuracy and efficiency in processing of Chinese texts. We also validated and expanded the model's application within the medical domain. Considering the characteristics of whole-course management, such as multidisciplinary collaboration and the involvement of multiple roles, we designed and developed an ontology for hypertensive cerebral hemorrhage applicable to whole-course management. We standardized the definition of its knowledge framework by referring to existing ontologies. In the knowledge collection phase, web scraping techniques were employed for the automated collection of partial data. In the knowledge-extraction phase, the predefined ontology model was optimized and a small amount of annotated data was used to fine-tune the ERNIE-UIE model, thereby achieving efficient joint extraction of entities and relationships using the pretrained language model. Finally, predefined mapping rules were used for data mapping and verification, ensuring the validity of the knowledge triples extracted.

In future work, we plan to attempt to optimize the ERNIE-UIE model or provide more standardized annotated datasets to observe changes in fine-tuning effects and compare the model's performance with those of other similar methods. In addition, we plan to conduct entity disambiguation for triples extracted.

## Supporting information

**S1 Appendix. Code for parsing knowledge extraction results.**
(DOCX)

**S2 Appendix. Code for mapping knowledge extraction results.**
(DOCX)

## Acknowledgments

We are grateful to the professional editors at Elsevier for language proofreading.

## Author contributions

**Conceptualization:** Bei Li, Jing Zheng.

**Writing – original draft:** Bei Li, Changbiao Li.

**Writing – review & editing:** Bei Li, Changbiao Li, Jianwei Sun, Xu Zeng, Xiaofan Chen, Jing Zheng.

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
