## [Decision Letter · Decision Letter 0]

17 Dec 2024

PONE-D-24-28816ERNIE-UIE: Advancing Information Extraction in Chinese Medical Knowledge GraphPLOS ONE

Dear Dr. zheng,

Thank you for submitting your manuscript to PLOS ONE. After careful consideration, we feel that it has merit but does not fully meet PLOS ONE’s publication criteria as it currently stands. Therefore, we invite you to submit a revised version of the manuscript that addresses the points raised during the review process.

We look forward to receiving your revised manuscript.

Kind regards,

Emiliano Damian Alvarez Leites, Ph.D.

Academic Editor

PLOS ONE

Journal Requirements:

2. In your Methods section, please include additional information about your dataset and ensure that you have included a statement specifying whether the collection and analysis method complied with the terms and conditions for the source of the data.

Reviewers' comments:

Reviewer's Responses to Questions

**Comments to the Author**

1. Is the manuscript technically sound, and do the data support the conclusions?

Reviewer #1: Partly

Reviewer #2: Partly

2. Has the statistical analysis been performed appropriately and rigorously? 

Reviewer #1: I Don't Know

Reviewer #2: N/A

3. Have the authors made all data underlying the findings in their manuscript fully available?

Reviewer #1: No

Reviewer #2: No

4. Is the manuscript presented in an intelligible fashion and written in standard English?

Reviewer #1: No

Reviewer #2: Yes

5. Review Comments to the Author

Reviewer #1: This paper explores the application of Universal Information Extraction (UIE) technology, specifically the ERNIE-UIE model, to construct a Chinese medical knowledge graph with minimal annotated data. By integrating ontology modeling, web scraping, and fine-tuning strategies, the study constructs a knowledge graph containing 8,525 entities and 9,522 relationships, which are verified using graph algorithms. The findings demonstrate that this approach effectively addresses the challenge of low-resource knowledge graph construction and enhances efficiency in knowledge extraction and storage.

My specialty is Knowledge Graphs (KGs), so I'm going to focus on that.

What the authors claim is a KG, is simply not that. They have a property graph, stored in Neo4J. A Knowledge Graph is a dataset constructed using triples represented in the RDF format (a W3C standard), which are annotated with a RDFS or OWL ontology (both are W3C standards), and are retrievable using the SPARQL query language (another W3C standard). Property graphs are not that.

The authors do not show any ontology being used, they do not show SPARQL queries being used, etc. They treat their constructed "knowledge graph" as a regular graph, using analytics which are common for graphs, not RDF knowledge graphs.

Additionally, the authors continuously use the term "triplets", which is a medical term for describing three siblings being born by the same mother at the same time. What they think about are structures called a "triples", e.g. RDF triples. A triple is a tuple of size three (3-tuple, triple).

Ultimately, the authors fail to show any meaningful reason why they converted the data into a Neo4J graph structure, when they could've simply loaded the parsed data into a relational database or a Python data structure and do the analysis they set out to do.

Reviewer #2: This paper proposes a methodology for constructing a medical knowledge graph by integrating ontology modeling, web scraping, Unified Information Extraction (UIE), fine-tuning strategies, and graph databases. It addresses knowledge modeling, extraction, and storage techniques. The ERNIE-UIE model was fine-tuned with a small amount of annotated data, enabling the joint extraction of entities and relationships. The resulting knowledge graph comprises 8,525 entity data points and 9,522 triplet data points, with its accuracy validated using graph algorithms. The paper demonstrates that adopting a generative extraction paradigm optimizes the construction process of Chinese medical knowledge graphs, achieving commendable results in low-resource settings and offering cost-efficient solutions for knowledge graph development.

Remaining Issues:

The paper lacks a detailed explanation of the data sources used for web scraping and annotation. Clarifying the data selection and processing methods would enhance reproducibility and credibility.

The paper provides a general description of the potential applications of the knowledge graph. It would benefit from specific examples of its use in medical scenarios to better demonstrate its practical value.

The paper does not discuss potential limitations of the generative extraction paradigm, such as its performance in handling complex entity relationships or its adaptability to other domains. Adding this discussion would provide a more balanced perspective.

Missing references:

Large Language Models for Generative Information Extraction: A Survey

LLMs for Knowledge Graph Construction and Reasoning: Recent Capabilities and Future Opportunities

6. PLOS authors have the option to publish the peer review history of their article (what does this mean? ). If published, this will include your full peer review and any attached files.

**Do you want your identity to be public for this peer review?** For information about this choice, including consent withdrawal, please see our Privacy Policy .

Reviewer #1: No

Reviewer #2: No

---

## [Author Response · Author response to Decision Letter 1]

20 Jan 2025

Journal Requirements:

Response:

We will check carefully to meet the journal's formatting requirements.

2. In your Methods section, please include additional information about your dataset and ensure that you have included a statement specifying whether the collection and analysis method complied with the terms and conditions for the source of the data.

Response:

The data sources statement was added in the methods section of the paper(See 3.1 Overall Framework), and a data availability statement was added at the end of the paper.

All data used in this study were obtained from publicly available sources. Web scraping was conducted strictly in accordance with the robots.txt protocols of the respective websites, ensuring compliance with their terms of service. Literature-derived data were exclusively sourced from open-access publications and public domain resources. The data collection and analysis methods fully complied with the terms and conditions of the original data sources. Careful attention was paid to respecting intellectual property rights and avoiding any potential copyright infringements. No proprietary or restricted-access data were utilized in this research. The data acquisition and use processes were designed to maintain ethical standards and legal requirements for scientific research.

Data Availability Statement

The authors confirm that the data supporting the findings of this study are available within the article and its supplementary materials.

Response:

We have supplemented the information about the funding and indicated it in the cover letter.

Reviewer #1: This paper explores the application of Universal Information Extraction (UIE) technology, specifically the ERNIE-UIE model, to construct a Chinese medical knowledge graph with minimal annotated data. By integrating ontology modeling, web scraping, and fine-tuning strategies, the study constructs a knowledge graph containing 8,525 entities and 9,522 relationships, which are verified using graph algorithms. The findings demonstrate that this approach effectively addresses the challenge of low-resource knowledge graph construction and enhances efficiency in knowledge extraction and storage.

Response:

Thanks for your positive comments. We have carefully revised this manuscript based on the reviewer’s comments. We hope that the revisions and improvements would satisfactorily address the reviewer’s concerns.

1.What the authors claim is a KG, is simply not that. They have a property graph, stored in Neo4J. A Knowledge Graph is a dataset constructed using triples represented in the RDF format (a W3C standard), which are annotated with a RDFS or OWL ontology (both are W3C standards), and are retrievable using the SPARQL query language (another W3C standard). Property graphs are not that.

Response:

Thanks for the valuable comments. We carefully searched for relevant information and explained the reasons why neo4j was used.

Reference:

https://neo4j.com/blog/rdf-vs-property-graphs-knowledge-graphs/

RDF-based storage and graph database-based storage are two common knowledge graph storage methods. RDF-based storage uses RDF format to represent triple information, which has the characteristics of flexibility, openness, and semantic expression. The graph database-based storage stores entities and relationships in a graph structure, which has the characteristics of association, fast query, elastic expansion, and complex analysis. In practical applications, the selection of a suitable storage method needs to consider specific application scenarios and requirements. With the development of knowledge graphs, it is believed that these two storage methods will continue to be widely used and bring more convenience and value to the construction and application of knowledge graphs.

Currently neo4j has been widely used in knowledge graphs.

Added References

[14]Yuan D, Zhou K, Yang C. Architecture and Application of Traffic Safety Management Knowledge Graph Based on Neo4j[J]. Sustainability, 2023, 15(12): 9786.

[15]Smith K, Liu F, Phanish D, et al. Research Collaboration Discovery through Neo4j Knowledge Graph[M]//Practice and Experience in Advanced Research Computing 2024: Human Powered Computing. 2024: 1-7.

(See Reference14,15)

Therefore, we added the following relevant content in the paper.

Neo4j (a graph database) was the knowledge storage tool selected for this study. It is a high-performance graph database management system that employs graph theory-based storage structures to manage data. Developed by Neo Technology, it is one of the most popular graph databases on the market, particularly well-suited for handling highly interconnected data. Based on the graph data model, Neo4j’s data structure consists of four components: nodes, relationships, properties, and labels. Compared with RDF storage, Neo4j offers significant advantages in terms of graph traversal performance, intuitive data modeling, real-time querying, and tool ecosystem, making it ideal for application scenarios requiring efficient graph operations and real-time responsiveness[14,15].

(see 4.3 Knowledge Storage of Hypertensive Intracerebral Hemorrhage)

2.The authors do not show any ontology being used, they do not show SPARQL queries being used, etc. They treat their constructed "knowledge graph" as a regular graph, using analytics which are common for graphs, not RDF knowledge graphs.

Response:

We sincerely appreciate the valuable comments. We have checked the literature carefully, and cited the related content as following.

Reference:

Khan A. Knowledge graphs querying[J]. ACM SIGMOD Record, 2023, 52(2): 18-29.

A number of technologies exist for KG querying, e.g.,SPARQL, SQL extensions, Datalog, graph query lan guages, keyword query, exampler query, faceted search,visual query, query templates, natural language ques tions, conversational QA, multimodal QA, and interac tive methods (e.g., feedback, explanation, suggestion,autocompletion, etc.).SPARQL is the W3C recommended query language for RDF. Neo4j provides a native graph database with property graph data model and Cypher query language.It also supports the Apache TinkerPop acting as a connectivity layer to use Grem lin. Neo4J supports several graph analytic tools (e.g.,Popoto.js, Neo4j Bloom) that assist in interactive, visual query building and suggestion. Neo4J’s graph data science library implements three graph embedding methods (FastRP, GraphSAGE, and Node2Vec), node classification and regression, link prediction.The top-10 commercial graph databases support various languages for querying of KGs – as RDF triples or property graphs. (Details in the following figure)

Therefore, our answer to this point is that, Cyper is more appropriate than SPARQL based on the database technology used in this study.

3.Additionally, the authors continuously use the term "triplets", which is a medical term for describing three siblings being born by the same mother at the same time. What they think about are structures called a "triples", e.g. RDF triples. A triple is a tuple of size three (3-tuple, triple).

Response:

We sincerely thank the reviewer for careful reading.

As suggested by the reviewer, we have corrected the “triplets” into “triple”.

4.Ultimately, the authors fail to show any meaningful reason why they converted the data into a Neo4j graph structure, when they could've simply loaded the parsed data into a relational database or a Python data structure and do the analysis they set out to do.

Response:

Thanks for the valuable comments.Based the comments, we have carefully reviewed some articles,and cited the related content as following.

Reference:

Vicknair C, Macias M, Zhao Z, et al. A comparison of a graph database and a relational database: a data provenance perspective[C]//Proceedings of the 48th annual Southeast regional conference. 2010: 1-6.

“Both systems performed acceptably on the objective benchmark tests. In general, the graph database did better at the structural type queries than the relational database. In full-text character searches, the graph databses performed significantly better than the relational database.”

Neo4j has advantages over RDF storage or SQL and other relational database storage.The advantage of graph databases is that they can naturally represent knowledge graph structures. The nodes in the graph represent objects in the knowledge graph, and the edges in the graph represent object relationships in the knowledge graph. The advantage of this approach is that the database itself provides a complete graph query language and supports various graph mining algorithms. The query speed is better than that of relational databases, especially the performance of multi-hop queries.

Therefore, our response to this comment is to add a description of the advantages of neo4j.

Compared with RDF storage, Neo4j offers significant advantages in terms of graph traversal performance, intuitive data modeling, real-time querying, and tool ecosystem, making it ideal for application scenarios requiring efficient graph operations and real-time responsiveness[14,15].

(See the 4.3 Knowledge Storage of Hypertensive Intracerebral Hemorrhage)

Reviewer #2: This paper proposes a methodology for constructing a medical knowledge graph by integrating ontology modeling, web scraping, Unified Information Extraction (UIE), fine-tuning strategies, and graph databases. It addresses knowledge modeling, extraction, and storage techniques. The ERNIE-UIE model was fine-tuned with a small amount of annotated data, enabling the joint extraction of entities and relationships. The resulting knowledge graph comprises 8,525 entity data points and 9,522 triplet data points, with its accuracy validated using graph algorithms. The paper demonstrates that adopting a generative extraction paradigm optimizes the construction process of Chinese medical knowledge graphs, achieving commendable results in low-resource settings and offering cost-efficient solutions for knowledge graph development.

Response:

We feel great thanks for your professional review work on our article. As you are concerned, there are several problems that need to be addressed.According to your nice suggestions, we have made extensive corrections to our previous draft, the detailed correction are listed below. 

1.The paper lacks a detailed explanation of the data sources used for web scraping and annotation. Clarifying the data selection and processing methods would enhance reproducibility and credibility.

Response:

Thanks for the valuable comments.The ontology construction process in this article refers to the existing OMAHA shcema in China and the Hypertension Ontology in the Open Bioinformatics Ontology (OBO) system.

Therefore, our response to this comment was to add a description of the data source in the paper.

Using existing ontologies such as the Omaha Schema and the hypertension ontology within the Open Biological and Biomedical Ontology framework as a reference, an ontology model for hypertensive intracerebral hemorrhage was constructed, based on which the entity and relationship types were determined, following which the text data were labeled.

See 3.1 Overall Framework)

The data sources used in the study were obtained via crawling the question-and-answer data from the professional medical Q&A website Dingxiang Doctor, combined with raw text data from professional literature and books sourced from China National Knowledge Infrastructure, Wanfang Medical Network, and WeChat Reading, totaling 261,494 characters.

See 3.2 Data Preprocessing

2.The paper provides a general description of the potential applications of the knowledge graph. It would benefit from specific examples of its use in medical scenarios to better demonstrate its practical value.

Response:

Thanks for the valuable comments.

Therefore, our response to this comment was to add example of using UIE technology to build knowledge graphs for public health emergency management.

This method holds certain potential for application in the medical field. Drawing on our previous research[18], if it is applied to knowledge extraction in the medical domain, it can provide a reliable source of medical knowledge for medical question answering and the currently popular retrieval-augmented generation research.

See Discussion)

3.The paper does not discuss potential limitations of the generative extraction paradigm, such as its performance in handling complex entity relationships or its adaptability to other domains. Adding this discussion would provide a more balanced perspective.

Response:

We sincerely appreciate the valuable comments.

We have checked the literature carefully and added morereferences into the Discussion part in the revised manuscript.

Although the generative extraction paradigm utilized in this study demonstrated significant potential in knowledge graph construction and IE, its limitations cannot be ignored, such as its heavy reliance on the design of prompts for performance. Future research needs to address various aspects, including prompt engineering, knowledge updating, hallucination issues, complex-sentence processing, domain adaptability, multitask learning, computational resource requirements, entity disambiguation, few-shot learning, and interpretability, to further enhance the performance and practicality of the generative extraction paradigm. By integration of external knowledge sources, introduction of postprocessing steps, and designing of more sophisticated multitask learning frameworks, among other technical approaches, the generative extraction paradigm is poised to play an even greater role in the field of knowledge graphs[19,20].

(See Discussion)

---

## [Decision Letter · Decision Letter 1]

1 Apr 2025

PONE-D-24-28816R1ERNIE-UIE: advancing information extraction in Chinese medical knowledge graphPLOS ONE

Dear Dr. zheng,

Thank you for submitting your manuscript to PLOS ONE. After careful consideration, we feel that it has merit but does not fully meet PLOS ONE’s publication criteria as it currently stands. Therefore, we invite you to submit a revised version of the manuscript that addresses the points raised during the review process.

We look forward to receiving your revised manuscript.

Kind regards,

Emiliano Damian Alvarez Leites, Ph.D.

Academic Editor

PLOS ONE

Journal Requirements:

Additional Editor Comments:

Comments from PLOS Editorial Office: We note that one or more reviewers has recommended that you cite specific previously published works. As always, we recommend that you please review and evaluate the requested works to determine whether they are relevant and should be cited. It is not a requirement to cite these works. We appreciate your attention to this request.

Reviewers' comments:

Reviewer's Responses to Questions

**Comments to the Author**

1. If the authors have adequately addressed your comments raised in a previous round of review and you feel that this manuscript is now acceptable for publication, you may indicate that here to bypass the “Comments to the Author” section, enter your conflict of interest statement in the “Confidential to Editor” section, and submit your "Accept" recommendation.

Reviewer #2: All comments have been addressed

Reviewer #3: (No Response)

Reviewer #4: (No Response)

2. Is the manuscript technically sound, and do the data support the conclusions?

Reviewer #2: Yes

Reviewer #3: Yes

Reviewer #4: Yes

3. Has the statistical analysis been performed appropriately and rigorously? 

Reviewer #2: Yes

Reviewer #3: Yes

Reviewer #4: Yes

4. Have the authors made all data underlying the findings in their manuscript fully available?

Reviewer #2: Yes

Reviewer #3: Yes

Reviewer #4: Yes

5. Is the manuscript presented in an intelligible fashion and written in standard English?

Reviewer #2: Yes

Reviewer #3: Yes

Reviewer #4: Yes

6. Review Comments to the Author

Reviewer #2: The revised version has addressed my concerns.I suggest adding a discussion on the latest works in large language model-based information extraction to enhance the paper's insights. For example:

Large language models for generative information extraction: A survey

Structured information extraction from scientific text with large language models

Revisiting relation extraction in the era of large language models

IEPile: unearthing large scale schema-conditioned information extraction corpus

Autore: Document-level relation extraction with large language models

Reviewer #3: This paper presents ERNIE-UIE, a method for extracting information from Chinese medical texts to build knowledge graphs with minimal annotated data. The research team fine-tuned the ERNIE-UIE model using just 200 annotated entries to extract entities and relationships from medical texts about hypertensive intracerebral hemorrhage.

The paper would benefit from a more detailed technical description of the prompt construction process for ERNIE-UIE. While the authors mention that prompts follow the format of "head entity's relationship type" and briefly explain the Structural Schema Instructors (SSI) concept, it may be helpful to provide sufficient details about prompt optimization for the medical domain.

It would be useful to state what specific procedures were used to handle medical entity disambiguation during graph construction, and how the authors addressed connectivity issues in the knowledge graph.

The paper would greatly benefit from including a discussion of recent large language model (LLM)-based information extraction approaches for knowledge graph construction. The authors should expand their literature review to include cutting-edge techniques like GraphRAG and LightRAG, which represent significant advancements in combining retrieval-augmented generation with graph-based knowledge structures.

Reviewer #4: (No Response)

7. PLOS authors have the option to publish the peer review history of their article (what does this mean? ). If published, this will include your full peer review and any attached files.

**Do you want your identity to be public for this peer review?** For information about this choice, including consent withdrawal, please see our Privacy Policy .

Reviewer #2: No

Reviewer #3: **Yes: ** Chao Huang

Reviewer #4: No

---

## [Author Response · Author response to Decision Letter 2]

22 Apr 2025

Journal Requirements:

1.Please review your reference list to ensure that it is complete and correct. If you have cited papers that have been retracted, please include the rationale for doing so in the manuscript text, or remove these references and replace them with relevant current references. Any changes to the reference list should be mentioned in the rebuttal letter that accompanies your revised manuscript. If you need to cite a retracted article, indicate the article’s retracted status in the References list and also include a citation and full reference for the retraction notice.

Response:

In accordance with the editor’s suggestions, we carefully reviewed all cited references to ensure that none of the cited works have been retracted.

2.Comments from PLOS Editorial Office: We note that one or more reviewers has recommended that you cite specific previously published works. As always, we recommend that you please review and evaluate the requested works to determine whether they are relevant and should be cited. It is not a requirement to cite these works. We appreciate your attention to this request.

Response:

We carefully reviewed all references suggested by the reviewers and selectively incorporated those most relevant to the content of the manuscript, in order to ensure academic rigor and the quality of citations.

Reviewer #2: The revised version has addressed my concerns.I suggest adding a discussion on the latest works in large language model-based information extraction to enhance the paper's insights. For example:

Large language models for generative information extraction: A survey

Structured information extraction from scientific text with large language models

Revisiting relation extraction in the era of large language models

IEPile: unearthing large scale schema-conditioned information extraction corpus

Autore: Document-level relation extraction with large language models

Response:

Thanks for your positive comments. We have extended the Introduction section to address the evolving role of large language models in information extraction, highlighting both their potential and limitations.

With the rapid advancement of large language models (LLMs), the landscape of information extraction is undergoing a profound transformation. Unlike traditional approaches based on classification or sequence labeling, LLMs have ushered in a new paradigm known as generative information extraction, wherein extraction tasks are reformulated as text generation problems. This paradigm enables the model to produce structured outputs—such as knowledge triples—in natural language form[14].Wadhwa et al. suggest that in the era of LLMs, relation extraction is shifting from typological modeling toward open-domain, question-answering-based modeling. By leveraging natural language prompts, this approach enhances scalability and cross-domain transferability[15].Nevertheless, generative extraction approaches continue to grapple with challenges such as inconsistent outputs and the omission of multi-entity or complex relational structures. To address these limitations, recent studies have advocated for the use of predefined schemas or domain ontologies to guide model outputs, thereby enforcing consistency and completeness. For instance, schema-conditioned extraction frameworks utilize structured templates to constrain and inform generation, offering a promising pathway toward building high-quality, domain-specific knowledge graphs[16].

see Introduction line 96-110 )

Reviewer #3: This paper presents ERNIE-UIE, a method for extracting information from Chinese medical texts to build knowledge graphs with minimal annotated data. The research team fine-tuned the ERNIE-UIE model using just 200 annotated entries to extract entities and relationships from medical texts about hypertensive intracerebral hemorrhage.

1.The paper would benefit from a more detailed technical description of the prompt construction process for ERNIE-UIE. While the authors mention that prompts follow the format of "head entity's relationship type" and briefly explain the Structural Schema Instructors (SSI) concept, it may be helpful to provide sufficient details about prompt optimization for the medical domain.

Response:

Thanks for the valuable comments.

The corresponding details are presented in both 3.2 data preprocessing section (line 222-236)and Section 4.2 Joint Entity-Relationship Extraction of the manuscript(358-389).

2.It would be useful to state what specific procedures were used to handle medical entity disambiguation during graph construction, and how the authors addressed connectivity issues in the knowledge graph.

Response:

Thanks for the valuable comments.We have acknowledged in the Discussion section that a limitation of this study is the lack of entity disambiguation, which remains an important issue to be addressed in future work. We sincerely appreciate the reviewer’s insightful comments regarding this point.

However, the model poorly performed for certain user-defined entity types and relationship types (see Tables 3 and 4). This may be because it is a pretrained language model, because of which although it retains the capability to extract general information from large-scale data during pretraining, it is not ideal for extraction of long phrases or sentences that lack typical sentence patterns and grammatical features. After fine-tuning, some metrics continued to exhibit poor performance and there was no entity disambiguation for the knowledge extracted, resulting in some semantically similar entities within the final knowledge set.

see Discussion line 542-549)

In response, additional content has been incorporated to clarify how connectivity challenges within the knowledge graph were resolved.

In this study, the ERNIE-UIE model was employed for knowledge extraction, and several key characteristics of the model are noteworthy. First, the model adopts a prompt-based strategy for triple extraction, in which the prompt serves as the input and is constructed using the head entity and the relation in a natural language-compliant format, such as "A's B" or the genitive form "B of A"—for instance, "intervention for disease." This design enables the generated prompt to align closely with linguistic intuitiveness and semantic clarity.Second, the output of the model—formatted as a JSON file—contains detailed information about the head entity (including its content and entity type), the relation type, and the tail entity content. However, it lacks information regarding the entity type of the tail entity.To address these issues, the semantic compatibility between the head entity and the relation was carefully considered during prompt construction, particularly in terms of the noun–verb collocational semantics, to ensure that the generated prompts retain meaningful interpretation and thereby enhance extraction performance. In addition, to compensate for the absence of tail entity type annotations, a strategy based on unique relation types was implemented. This approach allows for the post hoc inference of the tail entity type by mapping the extracted results according to predefined head–relation–tail structure patterns, wherein each unique relation reliably corresponds to a specific tail entity type.

(see 4.2 Joint Entity-Relationship Extraction line 329-344)

3.The authors should expand their literature review to include cutting-edge techniques like GraphRAG and LightRAG, which represent significant advancements in combining retrieval-augmented generation with graph-based knowledge structures.

Response:

We appreciate your valuable suggestion. In response, we have incorporated additional discussion on research related to retrieval-augmented generation (RAG).

Notably, in recent natural language processing (NLP) research, retrieval-augmented generation (RAG) has emerged as a widely adopted framework that integrates generative language models with external knowledge retrieval. This approach has demonstrated considerable utility across a range of complex tasks, including multi-hop question answering, information extraction, and knowledge graph construction. As the complexity of application scenarios continues to grow, research within the RAG paradigm has begun to shift from unstructured document retrieval toward more structured, controllable, and interpretable forms of knowledge integration. Of particular interest is the incorporation of graph-structured knowledge, as exemplified by recent frameworks such as GraphRAG and LightRAG.GraphRAG represents a notable advancement in this area by combining retrieval-augmented generation with knowledge graph reasoning. It transforms retrieved documents into entity-relation graphs and applies graph neural networks (GNNs) to enhance the model’s capacity for structural semantic understanding and logical inference during text generation. Empirical evidence has shown that this method significantly improves answer consistency and factual accuracy on challenging multi-hop tasks such as HotpotQA [27].

However, GraphRAG faces practical challenges, including high graph construction overhead and deployment complexity. In response, lightweight alternatives such as LightRAG have been introduced. LightRAG simplifies the graph structure into shallow entity connectivity graphs and incorporates efficient retrieval indexing techniques, thereby maintaining comparable performance while substantially reducing inference latency and resource consumption [28].To further advance the integration of generative models with structured information, recent studies have developed resources such as IEPile—a schema-conditioned information extraction corpus designed to support transfer learning and capability evaluation of large language models in structured output tasks across multiple domains[16].

In addition, recent studies have explored the incorporation of graph-structured constraints directly into the decoding process of retrieval-augmented generation (RAG) frameworks. Recent studies have explored the direct incorporation of structural graph constraints into the decoding process of retrieval-augmented generation (RAG) frameworks. For instance, GraphRAFT—a retrieval and reasoning framework—fine-tunes large language models (LLMs) to generate verifiably correct Cypher queries, thereby retrieving high-quality subgraph contexts and enabling the production of accurate, evidence-based answers. This approach leverages the synergy between structured knowledge retrieval and generative reasoning, addressing both the reliability and precision of knowledge-intensive outputs[29].In medical settings, certain approaches have sought to integrate specialized knowledge graphs to augment the performance of large language models (LLMs). For example, MedGraphRAG leverages a medical knowledge graph—customized specifically for processing private healthcare data—to enhance the capabilities of LLMs[30].Collectively, the integration of graph-structured knowledge into RAG architectures has significantly enhanced the model’s capabilities in structural recognition, factual reasoning, and semantic consistency. Future research may further investigate adaptive knowledge graph construction, cross-task representation sharing of graph structures, and interpretable graph-based inference. These directions hold substantial promise for expanding the utility of graph-enhanced RAG models in large-scale question answering and knowledge extraction tasks.

(see Discussion line 560-601)

Reviewer #4:The authors performed an information extraction in Chinese medical texts based on ERNIE-UIE. They expanded their work and developed an ontology model for whole-course management. The scope of work is diverse and the topic is interesting. However, the work lacks sufficient depth, with inadequate review of the most recent literature and insufficient elaboration on the innovative aspects of the research.

1. The paper suffers from inadequate literature citations, with limited investigation into extraction methodologies outside the medical domain and insufficient critical discussion of the limitations inherent in the cited medical research. There are additional works in the existing literature that should be referenced, including but not limited to:

- Ning Pang, Xiang Zhao, Weixin Zeng, Zhen Tan, Weidong Xiao. StaRS: Learning a Stable Representation Space for Continual Relation Classification. IEEE Transactions on Neural Networks and Learning Systems.

Response:

Thanks for the valuable comments.We acknowledge the shortcomings in our literature review and have accordingly supplemented this section to address the gaps.

In the field of materials science, Hei et al. proposed a novel framework integrating pointer networks and augmented attention mechanisms to extract complex multi-tuple relations from scientific literature, particularly those pertaining to alloy mechanical properties[11]. Their approach was designed to overcome the limitations of traditional methods in handling nested entities, contextual ambiguity, and one-to-many relational mappings. However, unlike materials science, medical informatics presents additional challenges, including higher annotation costs and a more dynamic data landscape—particularly in response to emerging disease variants. These characteristics necessitate the development of models with enhanced transfer learning capabilities.To address continual relation classification, Pang et al.introduced a strategy that combines knowledge distillation with contrastive learning to mitigate the problem of catastrophic forgetting. While effective in general domains, their method did not account for the unique temporal dynamics of medical data, such as the frequent updates to clinical guidelines and evolving diagnostic criteria [12].

see introduction line 66-78)

2. The paper critically lacks a clear articulation of research motivation in the Introduction section. The analysis of challenges specific to Chinese-language medical information extraction remains superficial, resulting in underdeveloped justification for the study's innovative value. Furthermore, the authors must explicitly delineate the technical distinctions between their methodology and the ERNIE-UIE framework in the Introduction; otherwise, this work risks being perceived as a straightforward implementation of ERNIE-UIE rather than a substantive methodological advancement.

Response:

Thanks for the valuable comments.To address this shortcoming, we have incorporated additional literature in the Introduction.

In the development of Chinese medical information extraction, scholars have made continuous progress while simultaneously uncovering a range of challenges unique to this domain. One major linguistic obstacle arises from the absence of natural word boundaries in the Chinese language, which frequently leads to incorrect segmentation of clinical terms during the tokenization stage. Such segmentation errors can propagate through downstream tasks—most notably named entity recognition—resulting in cascading inaccuracies. For example, the phrase “高血压病史三年” (a three-year history of hypertension) should be recognized as a single unit representing a medical condition. However, if it is incorrectly segmented into “高/血压/病史” (high / blood pressure / history), the semantic integrity of the expression is easily compromised.Moreover, in contrast to English-language biomedical text mining—which benefits from well-established ontologies such as UMLS and SNOMED CT—the Chinese context still largely relies on manually curated lexicons or domain-specific knowledge graphs, such as CMKG [13]. These resources often exhibit limitations in terms of coverage, connectivity, and the degree of structural formalization.In addition, the sensitive and regulated nature of medical data presents a further obstacle, significantly restricting access to high-quality, publicly available corpora and annotated samples. Consequently, Chinese medical information extraction faces multifaceted challenges spanning linguistic representation, knowledge modeling, and task-level robustness.

(see Introduction line 79-95)

Building upon the ERNIE-UIE framework, this study incorporates a schema grounded in established medical ontologies to address domain-specific challeng

---

## [Decision Letter · Decision Letter 2]

6 May 2025

ERNIE-UIE: advancing information extraction in Chinese medical knowledge graph

PONE-D-24-28816R2

Dear Dr. zheng,

We’re pleased to inform you that your manuscript has been judged scientifically suitable for publication and will be formally accepted for publication once it meets all outstanding technical requirements.

Kind regards,

Emiliano Damian Alvarez Leites, Ph.D.

Academic Editor

PLOS ONE

Additional Editor Comments (optional):

Reviewers' comments:

Reviewer's Responses to Questions

**Comments to the Author**

1. If the authors have adequately addressed your comments raised in a previous round of review and you feel that this manuscript is now acceptable for publication, you may indicate that here to bypass the “Comments to the Author” section, enter your conflict of interest statement in the “Confidential to Editor” section, and submit your "Accept" recommendation.

Reviewer #2: All comments have been addressed

Reviewer #4: All comments have been addressed

2. Is the manuscript technically sound, and do the data support the conclusions?

Reviewer #2: Yes

Reviewer #4: Yes

3. Has the statistical analysis been performed appropriately and rigorously? 

Reviewer #2: N/A

Reviewer #4: Yes

4. Have the authors made all data underlying the findings in their manuscript fully available?

Reviewer #2: No

Reviewer #4: (No Response)

5. Is the manuscript presented in an intelligible fashion and written in standard English?

Reviewer #2: Yes

Reviewer #4: Yes

6. Review Comments to the Author

Reviewer #2: The proposed model in this paper integrates ontology modeling, web scraping, UIE, fine-tuning strategies, and graph databases, thereby covering knowledge modeling, extraction, and storage techniques. The revised version has addressed my concerns.

Reviewer #4: (No Response)

7. PLOS authors have the option to publish the peer review history of their article (what does this mean? ). If published, this will include your full peer review and any attached files.

**Do you want your identity to be public for this peer review?** For information about this choice, including consent withdrawal, please see our Privacy Policy .

Reviewer #2: No

Reviewer #4: No

---

## [Editor Report · Acceptance letter]

PONE-D-24-28816R2

PLOS ONE

Dear Dr. Zheng,

I'm pleased to inform you that your manuscript has been deemed suitable for publication in PLOS ONE. Congratulations! Your manuscript is now being handed over to our production team.

Kind regards,

on behalf of

Dr. Emiliano Damian Alvarez Leites

Academic Editor

PLOS ONE